# Association of Trabecular Bone Score-Adjusted Fracture Risk Assessment Tool with Coronary Artery Calcification in Women

**DOI:** 10.3390/diagnostics12010178

**Published:** 2022-01-12

**Authors:** Tzyy-Ling Chuang, Yuh-Feng Wang, Malcolm Koo, Mei-Hua Chuang

**Affiliations:** 1Department of Nuclear Medicine, Dalin Tzu Chi Hospital, Buddhist Tzu Chi Medical Foundation, Chiayi 622401, Taiwan; b8601139@tmu.edu.tw (T.-L.C.); yuhfeng@gmail.com (Y.-F.W.); 2School of Medicine, Tzu Chi University, Hualien 970374, Taiwan; 3Graduate Institute of Long-Term Care, Tzu Chi University of Science and Technology, Hualien 970302, Taiwan; 4Faculty of Pharmacy, National Yang Ming Chiao Tung University, Taipei City 112304, Taiwan; 5Department of Pharmacology, School of Medicine, Tzu Chi University, Hualien 970374, Taiwan

**Keywords:** coronary artery calcification, TBS-adjusted FRAX, trabecular bone score, adult women

## Abstract

The trabecular bone score (TBS) was found to be significantly associated with moderate coronary artery calcification (CAC). The aim of this study was to further explore the association between TBS-adjusted Fracture Risk Assessment Tool (FRAX) and CAC score in women. The electronic medical record database of a regional teaching hospital in southern Taiwan yielded women who received both coronary computed tomography and bone mineral density (BMD) measurement during their general health examination. Health history, anthropomorphic measurements, laboratory results, BMD, and T-scores were obtained. TBS values were calculated from database spine dual-energy X-ray absorptiometry files. Linear regression analyses tested the association between CAC score and 10-year probability of major osteoporotic fracture (MOF) and hip fracture (HF) determined by TBS-adjusted FRAX. Of the 116 women (mean age 55.8 years) studied, 24.1% had osteoporosis. Simple linear regression showed a significant association of CAC score with an increase in MOF and HF risk as measured by TBS-adjusted FRAX. In multiple linear regression adjusted for potential confounders, CAC score remained significantly associated with TBS-adjusted FRAX for right MOF (*p* = 0.002), left MOF (*p* = 0 006), right HF (*p* = 0.005), and left HF (*p* = 0.015). In conclusion, clinicians should be vigilant to the potential increased risk of coronary events among women with increased TBS-adjusted FRAX for MOF and HF.

## 1. Introduction

Vascular calcification is closely related to vascular injury and inflammation [1]. Coronary artery calcification (CAC) can independently predict the presence of coronary atherosclerotic plaque [2] and future cardiac events [3]. A cohort study of 10,377 asymptomatic individuals found that CAC was a strong predictor of morality, and it could significantly improve outcome classification compared with the Framingham risk score [4]. Another study on 2028 asymptomatic older adults also indicated that CAC scoring was a useful method to reclassify individuals into appropriate risk categories [5]. A population-based study of 6722 men and women also showed that the CAC score was a strong predictor of incident coronary events in various ethnic groups, including white, black, Chinese, and Hispanic adults in the United States [6]. According to the 2018 American Heart Association and American College of Cardiology (AHA/ACC) cholesterol management guideline, CAC testing is encouraged to implement shared decision making and to individualize treatment plans, particularly for primary atherosclerotic cardiovascular disease prevention in asymptomatic patients and those with borderline and intermediate risk [7].

Trabecular bone score (TBS) is a texture parameter that quantifies local variation in the gray level distribution of anteroposterior DXA images [8,9] of the lumber spine. Although not a direct physical measure of bone microarchitecture, TBS significantly correlates with three-dimensional parameters of bone microarchitecture independently of areal BMD [10]. A higher TBS indicates a stronger and more fracture-resistant microarchitecture. Unlike quantitative computed tomography, with its higher radiation exposure and greater expense, TBS is less expensive and requires only additional software analysis of lumbar spine BMD [8]. In our previous study, we found that TBS was significantly associated with moderate CAC in patients [11] and that the fracture risk assessment tool (FRAX) was significantly and independently associated with the CAC score [12]. However, the association between TBS-adjusted FRAX and CAC has not yet been explored. Therefore, the association between TBS-adjusted FRAX and CAC score were investigated in this study of adult women.

## 2. Materials and Methods

### 2.1. Subjects and Study Variables

In this retrospective medical record review study, women aged 20 to 80 years who had received both coronary computed tomography angiography and DXA scans at their general health examination from June 2014 to July 2020 at a regional teaching hospital in southern Taiwan were reviewed. Exclusion criteria included the presence of metal implants in the measured body parts or the possibility of pregnancy.

The study protocol was approved by the institutional review board of Dalin Tzu Chi Hospital (IRB No. B11001010), which waived the requirement for obtaining informed consent from patients.

The following information was ascertained from the medical records of the participants: (1) anthropometric characteristics, including age, height, and weight; (2) comorbidities, including hypertension, diabetes mellitus, and hyperlipidemia; (3) laboratory findings, including systolic blood pressure, diastolic blood pressure, high-density lipoprotein cholesterol, low-density lipoprotein cholesterol, total cholesterol, triglycerides, fasting glucose, calcium, alkaline phosphatase, and estimated glomerular filtration rate; and (4) other potential confounders, including previous fracture history, parental history of fractured hip, smoking status, use of glucocorticoids, rheumatoid arthritis, secondary osteoporosis, and alcohol use. Smoking status and alcohol use were defined according to the FRAX tool. Smoking is defined as yes if the patient currently smokes tobacco. Alcohol use is defined as yes if the patient takes three or more units of alcohol daily. A unit of alcohol is approximately equal to a standard glass of beer (285 mL) or a single measure of spirits (30 mL). Blood sampling, coronary computed angiography, and DXA scans for every patient were performed within a single general health examination.

### 2.2. Measurement of Coronary Artery Calcification (CAC)

Unenhanced axial images (with tube voltage of 120 kVp, slice thickness at 2.5 mm, 512 matrix size, original thin-slice collimation, EKG-gating, breath holding) scanned before coronary computed angiography were used to obtain CAC scores. The scans were performed using a multidetector computed tomography system (LightSpeed VCT, GE Medical Systems, GE Healthcare, Chicago, IL, USA). The Agatston scoring method was used to quantify CAC [13]. Automated detection of calcific lesions was first processed on an Advantage Workstation, AW 4.3-09 (GE Healthcare, Chicago, IL, USA), and the exclusion of calcific lesions other than the coronary arteries (such as bones, lymph nodes, and lung lesions) was performed manually. A total calcium score was obtained by summing the individual scores of the left main, left anterior descending, circumflex, and right coronary arteries with their main branches.

### 2.3. Measurement of Bone Mineral Density

Bone mineral density (BMD) at the lumbar spine and bilateral hips (total and femoral neck regions) was measured using dual-energy X-ray absorptiometry (DXA), which was performed with a DiscoveryWi DXA system (Hologic Inc., Marlborough, MA, USA). Individuals whose BMD measured areas containing metal materials were excluded. Individuals were classified as having osteoporosis (T-score < −2.5 standard deviations [SD]) or osteopenia (T-score −1.0 to −2.5 SD) based on the World Health Organization classification.

### 2.4. Measurement of Trabecular Bone Score (TBS)

In this study, TBS was retrospectively quantified using TBS iNsight software, Version 3.0.2.0 (MedImaps, Geneva, Switzerland) on existing DXA scans of the spine of participants in our medical record database.

### 2.5. The Fracture Risk Assessment Tool (FRAX) and TBS-Adjusted FRAX Calculations

The 10-year probabilities (expressed as a percentage) of major osteoporotic fracture (MOF) and hip fracture (HF) were calculated. Fracture risk factors, including age, sex, weight, height, history of fracture, parental history of fractured hip, current smoking, use of glucocorticoids, rheumatoid arthritis, secondary causes of osteoporosis, and alcohol intake, were included in the FRAX calculation. The TBS-adjusted FRAX probabilities for HF and MOF were calculated using the country-specific (Taiwan) tool provided on the FRAX website (www.sheffield.ac.uk/FRAX/tool.aspx?lang=cht, Accessed date 11 January 2022).

### 2.6. Statistical Analysis

Summary statistics were expressed as mean and SD or number and percentage, as appropriate. Differences in means or frequencies of characteristics and CAC between the normal, osteopenia, and osteoporosis groups were evaluated using the chi-square test or analysis of variance, as appropriate. The Bonferroni correction was applied to adjust for multiple comparisons.

Simple linear regression analysis was performed with CAC scores as the dependent variable with clinical characteristics, laboratory data, and TBS-adjusted FRAX treated as the independent variables. Multiple linear regression analyses of CAC scores and TBS-adjusted FRAX were conducted, adjusting for age, hypertension, hyperlipidemia, and systolic blood pressure, and the independent variables that the simple linear regression analysis found were significantly associated with CAC scores. All statistical analyses were performed using PASW Statistics for Windows, Version 18.0 (SPSS Inc., Chicago, IL, USA).

## 3. Results

### 3.1. Characteristics of the Participants

A total of 116 women with a mean age of 55.4 years (SD 8.3, range 26–75 years) were included in the study. Based on the T-score for BMD, 26 of the participants were classified as normal (22.4%), 62 classified as having osteopenia (53.4%), and 28 as having osteoporosis (24.2%). Age was significantly older in women with osteopenia and osteoporosis as compared to those with normal BMD. Participants with osteoporosis had a lower body mass index than those with normal BMD or osteopenia. A significantly higher alkaline phosphatase level was observed in participants with osteoporosis, as compared with those with normal BMD. The overall mean CAC score was 34.7 ± 137.6 (range 0–1185), and no significant differences were observed between the three groups (Table 1).

### 3.2. Simple Linear Regression Analysis

#### 3.2.1. Factors Associated with CAC

Simple linear regression analysis showed that the CAC score was significantly correlated with age (standardized [std] β = 0.287, *p* = 0.002), hypertension (std β = 0.371, *p* < 0.001), hyperlipidemia (std β = 0.225, *p* = 0.015), and systolic blood pressure (std β = 0.220, *p* = 0.018). These four variables were subsequently included in the multiple regression models as potential confounders (Table 2).

#### 3.2.2. Simple Linear Regression Analysis of CAC

Simple linear regression analysis showed that TBS (std β = −0.214, *p* = 0.021) was significantly and inversely associated with CAC score. Furthermore, the TBS-adjusted FRAX at both the right and left MOF was significantly associated with the CAC score (std β = 0.383, *p* < 0.001; std β = 0.370, *p* < 0.001, respectively). Similarly, the TBS-adjusted FRAX at both the right and left HF was significantly associated with the CAC score (std β = 0.336, *p* < 0.001; std β = 0.309, *p* < 0.001, respectively) (Table 3).

### 3.3. Multiple Linear Regression Analysis of CAC

Results of the multiple linear regression analyses, with adjustments for age, hypertension, hyperlipidemia, and systolic blood pressure, showed that TBS was no longer significantly associated with CAC scores. In contrast, the TBS-adjusted FRAX for both the right and left MOF as well as for HF remained significantly associated with the CAC scores. The TBS-adjusted FRAX at both the right and left MOF were significantly associated with the CAC score (std β = 0.350, *p* = 0.002; std β = 0.298, *p* = 0.006, respectively). In addition, the TBS-adjusted FRAX at both the right and left HF were significantly associated with the CAC score (std β = 0.264, *p* = 0.005; std β = 0.225, *p* = 0.015, respectively) (Table 4).

## 4. Discussion

In this retrospective medical record review study of women who had undergone a general health examination, TBS-adjusted FRAX was significantly associated with CAC after adjustment for potential confounders. Our study also showed that age, hypertension, hyperlipidemia, and systolic blood pressure were significantly associated with the extent of CAC, a result which is consistent with previous research [12,14].

FRAX is widely used in clinical settings to assess the risk of fractures. The addition of TBS to FRAX might further improve its accuracy in predicting major osteoporotic fractures. In a study of 2012 community-dwelling older Japanese men, category-free integrated discrimination and net reclassification were significantly more accurate when the FRAX score was combined with TBS compared to FRAX alone [15]. Another study of 29,407 Canadian women also found that combining TBS with BMD could improve fracture prediction in postmenopausal women [16]. The findings of these studies support the use of TBS to adjust FRAX to increase its accuracy [17].

Nevertheless, studies in other populations have shown conflicting results. A community-based cohort study of 1165 Korean women found that FRAX with TBS adjustment did not show better predictive value for osteoporotic fractures than FRAX alone [18]. Another study of 358 postmenopausal Iranian women also reported that the addition of TBS to FRAX did not significantly improve the predictive value of vertebral fracture [19]. Further prospective cohort studies with long-term follow up are needed to explore the factors that contribute to the discrepancies between studies in different populations.

Previous studies showed that patients with cardiovascular disease were associated with a higher risk of fracture. A prospective cohort study of 31,936 Swedish twins revealed that a diagnosis of cardiovascular disease was significantly associated with risk of subsequent hip fracture [20]. Moreover, a study of 586 current and former smokers in the ECLIPSE cohort showed that the prevalence of vertebral fractures was significantly associated with the CAC score [21]. In our previous retrospective medical review study on 246 adult patients, no significant independent associations were observed between BMD of the lumbar spine, femoral neck, or total hip with a moderate or high CAC score in patients with osteopenia. In patients with osteoporosis, only the BMD of the lumbar spine was significantly and inversely associated with moderate CAC score [22]. In addition, we previously reported that TBS was independently associated with moderate but not high CAC scores [11]. In contrast, the finding of significant association between CAC scores and TBS-adjusted FRAX in the present study could be explained by the inclusion of other fracture risk factors in the calculation of FRAX and the increased predictive ability of TBS independent of FRAX clinical factors and BMD.

Osteoporosis and atherosclerosis are often present concomitantly in individuals [23]. Traditionally, these two conditions were considered age-related processes. However, a growing body of evidence suggested that the two conditions shared common pathophysiological mechanisms [24]. While the exact mechanism remains uncertain, several hypotheses have been proposed. Inhibitors of the Wnt signaling pathway, such as secreted frizzled Proteins 2 and 4 and Dickkopf-related protein-1, could play a role linking vascular calcification and bone loss [25]. Inflammatory cytokines and oxidized low-density lipoproteins have also been suggested as determinants of vascular calcification and decrease in osteoblast activity. Moreover, the RANKL/RANK/OPG system and the cysteine protease cathepsin K were hypothesized to regulate vascular calcification and bone metabolism [26].

There were several limitations to this study. First, data from this study were based on review of medical records of relatively healthy women who received a general health examination. Second, information on medications used and calcium intake was not available from the medical records of the general health examination. Third, laboratory clinical measurements, such as estrogen, vitamin D, thyroid hormone, parathyroid hormone, and osteoprotegerin, were not available to explore possible mechanisms linking CAC and bone fragility. Fourth, all data were obtained from a single regional hospital in southern Taiwan, which may limit the generalizability of the finding.

## 5. Conclusions

In this retrospective medical review study of those with an increased risk of MOF and HF, TBS-adjusted FRAX was significantly and independently associated with more severe CAC in female adults. TBS-adjusted FRAX could be used to predict both fracture risk and CAC severity. Early evaluation and treatment to reduce the risk of fracture and the risk of coronary events could be considered in women with high TBS-adjusted FRAX scores.

## Figures and Tables

**Table 1 diagnostics-12-00178-t001:** Demographic and clinical characteristics of the participants (*n* = 116).

Variable	Total *n* = 116 (100%)	T-Score Level			*p*
		Normal *n* = 26 (22.4%)	Osteopenia *n* = 62 (53.4%)	Osteoporosis *n* = 28 (24.2%)	
Age (years)	55.4 ± 8.3	51.2 ± 9.3 ^a^	56.0 ± 7.0 ^b^	57.8 ± 8.8 ^b^	0.008
Smoking (%)	1 (0.9)	0 (0.0)	1 (1.6)	0 (0.0)	>0.999
Alcohol use (%)	7 (6.0)	2 (7.7)	3 (4.8)	2 (7.1)	0.673
Hypertension (%)	24 (20.7)	8 (30.8)	12 (19.4)	4 (14.3)	0.305
Diabetes mellitus (%)	7 (6.0)	0 (0.0)	4 (6.5)	3 (10.7)	0.264
Hyperlipidemia (%)	8 (6.9)	1 (3.8)	6 (9.7)	1 (3.6)	0.625
Secondary osteoporosis (%)	21 (18.1)	4 (15.4)	11 (17.7)	6 (21.4)	0.854
Trabecular bone score	1.34 ± 0.11	1.43 ± 0.10 ^a^	1.35 ± 0.09 ^b^	1.26 ± 0.08 ^c^	<0.001
Right MOF (FRAX) (%)	6.4 ± 4.0	3.2 ± 1.7 ^a^	6.0 ± 3.2 ^b^	10.1 ± 4.3 ^c^	<0.001
Right HF (FRAX) (%)	1.8 ± 2.1	0.3 ± 0.3 ^a^	1.4 ± 1.4 ^b^	4.1 ± 2.5 ^c^	<0.001
Left MOF (FRAX) (%)	6.2 ± 4.2	3.1 ± 1.4 ^a^	5.8 ± 3.3 ^b^	10.1 ± 4.9 ^c^	<0.001
Left HF (FRAX) (%)	1.7 ± 2.3	0.2 ± 0.2 ^a^	1.3 ± 1.4 ^b^	4.1 ± 3.2 ^c^	<0.001
Body mass index (kg/m^2^)	24.5 ± 3.5	26.2 ± 4.3 ^a^	24.7 ± 2.9 ^a^	22.5 ± 2.8 ^b^	<0.001
Systolic blood pressure (mmHg)	124.6 ± 22.5	132.0 ± 23.3	123.6 ± 18.6	120.0 ± 28.2	0.126
Diastolic blood pressure (mmHg)	72.4 ± 13.0	75.7 ± 13.1	71.2 ± 10.1	71.9 ± 17.8	0.335
High-density lipoprotein (mg/dL)	54.5 ± 14.3	48.8 ± 10.1	55.6 ± 15.4	57.2 ± 14.2	0.064
Low-density lipoprotein (mg/dL)	124.1 ± 27.1	123.7 ± 27.4	123.0 ± 26.7	127.1 ± 28.5	0.794
Total cholesterol (mg/dL)	197.6 ± 31.7	192.9 ± 30.7	197.1 ± 31.2	203.1 ± 33.9	0.491
Triglycerides (mg/dL)	113.0 ± 63.4	132.5 ± 70.6	109.9 ± 64.8	101.9 ± 49.9	0.177
Glucose (mg/dL)	105.1 ± 20.6	106.2 ± 19.7	105.1 ± 19.4	104.1 ± 24.5	0.934
Calcium (mmol/L)	2.26 ± 0.11	2.26 ± 0.09	2.25 ± 0.08	2.27 ± 0.15	0.680
Alkaline phosphatase (IU/L)	75.6 ± 21.0	69.0 ± 18.6 ^a^	74.7 ± 18.1 ^ab^	83.6 ± 26.4 ^b^	0.032
eGFR (mL/min/1.73 m^2^)	110.7 ± 25.8	112.9 ± 28.1	107.7 ± 24.3	115.1 ± 27.0	0.406
CAC score	34.7 ± 137.6	30.9 ± 84.3	35.1 ± 163.2	37.4 ± 117.8	0.985

CAC, coronary artery calcification; eGFR, estimated glomerular filtration rate; FRAX, Fracture Risk Assessment Tool; HF, hip fracture; MOF, major osteoporotic fracture. Means in a row that do not share a common superscript differ significantly (*p* < 0.05, Bonferroni adjustment).

**Table 2 diagnostics-12-00178-t002:** Simple linear regression analysis of factors associated with the coronary artery calcification score in the participants.

Variable	β	95% Confidence Interval	Standardized β	*p*
Age (years)	4.77	1.81, 7.73	0.287	0.002
Body mass index (kg/m^2^)	−1.06	−8.39, 6.26	−0.027	0.774
Smoking	−20.68	−295.66, 254.30	−0.014	0.882
Alcohol use	−34.78	−141.35, 71.79	−0.060	0.519
Hypertension	125.61	67.34, 183.88	0.371	<0.001
Diabetes mellitus	85.49	−20.09, 191.07	0.149	0.111
Hyperlipidemia	121.81	24.06, 219.57	0.225	0.015
Secondary osteoporosis	14.71	−51.26, 80.68	0.041	0.660
Systolic blood pressure (mmHg)	1.34	0.24, 2.45	0.220	0.018
Diastolic blood pressure (mmHg)	1.68	−0.27, 3.62	0.158	0.090
High-density lipoproteins (mg/dL)	1.76	0.00, 3.51	0.183	0.050
Low-density lipoproteins (mg/dL)	−0.37	−1.31, 0.57	−0.074	0.433
Total cholesterol (mg/dL)	−0.08	−0.89, 0.73	−0.018	0.844
Triglycerides (mg/dL)	−0.16	−0.56, 0.24	−0.075	0.425
Fasting blood glucose (mg/dL)	0.29	−0.95, 1.53	0.044	0.641
Calcium (mmol/L)	−11.08	−254.15, 232.00	−0.008	0.928
Alkaline phosphatase (IU/L)	0.38	−0.83, 1.60	0.059	0.533
eGFR (mL/min/1.73 m^2^)	−0.50	−1.49, 0.48	−0.095	0.312

eGFR, estimated glomerular filtration rate.

**Table 3 diagnostics-12-00178-t003:** Simple linear regression analysis of bone parameters associated with the coronary artery calcification score in the participants.

Variable	β	95% Confidence Interval	Standardized β	*p*
TBS	−278.14	−513.94, −42.33	−0.214	0.021
Right MOF (TBS-adjusted FRAX) (%)	11.58	6.40, 16.76	0.383	<0.001
Right HF (TBS-adjusted FRAX) (%)	20.77	9.96, 31.57	0.336	<0.001
Left MOF (TBS-adjusted FRAX) (%)	10.64	5.69, 15.60	0.370	<0.001
Left HF (TBS-adjusted FRAX) (%)	17.15	7.35, 26.94	0.309	0.001

FRAX, Fracture Risk Assessment Tool; HF, hip fracture; MOF, major osteoporotic fracture; TBS, trabecular bone score.

**Table 4 diagnostics-12-00178-t004:** Multiple linear regression analysis of bone parameters associated with the coronary artery calcification score in the participants, adjusted for age, hypertension, hyperlipidemia, and systolic blood pressure.

Variable	Adjusted R^2^	β	95% Confidence Interval	Standardized β	*p*
TBS	0.211	−25.59	−290.92, 239.73	−0.020	0.849
Right MOF (TBS-adjusted FRAX) (%)	0.277	10.57	3.95, 17.19	0.350	0.002
Right HF (TBS-adjusted FRAX) (%)	0.265	16.35	4.92, 27.79	0.264	0.005
Left MOF (TBS-adjusted FRAX) (%)	0.264	8.56	2.53, 14.58	0.298	0.006
Left HF (TBS-adjusted FRAX) (%)	0.252	12.50	2.46, 22.54	0.225	0.015

FRAX, Fracture Risk Assessment Tool; HF, hip fracture; MOF, major osteoporotic fracture; TBS, trabecular bone score.

## Data Availability

The data used to support the findings of this study are available from the corresponding author upon request.

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
