# Peer review of "Association of Trabecular Bone Score-Adjusted Fracture Risk Assessment Tool with Coronary Artery Calcification in Women"

_diagnostics, 2022, doi:10.3390/diagnostics12010178_

Round 1

Reviewer 1 Report

The main research results, their theoretical and practical significance, the validity of the conclusions:

The results obtained indicate the effectiveness of the diagnostic technique, can be used not only in scientific terms, but also in practical health care. The conclusions of the work are substantiated.

Adequacy of the use of literary sources:

Literary sources are used adequately.

The quality of the article design (compliance with the editorial requirements, style, consistency, internal unity, language, etc.):

The article is well-prepared, the presentation style is good. There is a clear logical sequence of presentation of their own results.

Author Response

We thank the reviewer for taking the time and effort necessary to review our manuscript. 

Reviewer 2 Report

Summary

The authors investigated the trabecular bone score (TBS) and TBS-adjusted fracture risk assessment tool (FRAX) scores as predictors of coronary artery atherosclerosis imaging biomarkers. The authors performed a simple linear regression analysis to predict the coronary artery calcification (CAC) score based on clinical/laboratory/anthropometric variables as well as TBS and TBS-adjusted FRAX-scores as independent variables, respectively. The TBS and TBS-adjusted FRAX scores were further included into a multiple linear regression analysis as independent variables to predict CAC – the latter analysis was adjusted for age, hypertension, hyperlipidemia and systolic blood pressure (variables with a significant regression equation at the simple linear regression analysis). TBS-adjusted FRAX for right and left major osteoporotic- and hip fractures were significant predictors of CAC-score.

Strengths:

1. The article is well-written in proficient English.

2. Women with high TBS-adjusted FRAX score may benefit from cardiovascular screening which is potentially interesting for clinical practice.

Weaknesses:

While it is indeed relevant to gain knowledge on how FRAX predicts CAC, the importance of the TBS-adjusted FRAX is not highlighted in the manuscript. It is true that TBS-adjustment of FRAX may improve the fracture risk assessment but it remains unclear for the reader why it would be a better predictor of CAC compared to non-adjusted FRAX.

TBS-adjusted FRAX scores are especially useful for predicting fracture risk in postmenopausal women close to a FRAX-based intervention threshold (P Martineau, BC Silva, WD Leslie. Utility of trabecular bone score in the evaluation of osteoporosis. Curr Opin Endocrinol Diabetes Obes. 2017;24(6):402-410). The reader can assume that women outside this subpopulation were included in the current analysis which makes the clinical usefulness of the results questionable. Furthermore, FRAX is applied to women in the age range 40-90 years, but women with age outside this range have also been included in the current study.

Specific comments:

Title:

Consider using the full name instead of the abbreviation for FRAX.

Abstract:

The background and the objective of the study are not stated.

Page 1 line 29: “Increased risk of MOF and HF, determined by TBS-adjusted FRAX, is significantly and independently associated with CAC score in women.” – this is rather a repetition of the results that does not refer to the conclusion.

Keywords:

Page 1 line 31: “women” is a rather unspecific keyword, I recommend changing it.

Introduction:

The description of the pathophysiological link between coronary artery calcification and bone mineral density could be more detailed. It is also somewhat unclear why TBS-adjusted FRAX would be a better predictor of CAC compared to TBS and FRAX alone, which was previously investigated by the authors.

Materials and Methods:

The inclusion- and exclusion criteria are not described.

The details of the coronary artery computed tomography (CT) image acquisition are not described (resolution, kV, etc.). It is not stated which images were used for Agatston-score analysis (e.g. original thin-slice images or multiparametric reconstruction images). It is also unclear which software was used for Agatston-scoring.

The TBS and FRAX scores could be included in the descriptive analysis.

The time interval between the coronary artery CT and the dual-energy X-ray absorptiometry (DXA) scan is not described, which is important since longer times between the two scans could bias the results.

Page 2 line 64: the authors might mean “anthropometric” instead of “anthropomorphic” characteristics.

Page 2 line 65-69: It is unclear when the listed laboratory parameters were obtained in relation to the date of the DXA scan and the coronary CT scan.

Page 2 line 70-71: The authors do not describe the definition of smoking status and alcohol use.

Page 2 line 88-94: This description does not belong to the methods and would rather fit in the introduction and discussion.

Results

The words “associated” and “correlated” are frequently used in the results section. It is advised that the authors to also use “predicted/predictors” in order to adequately reflect the meaning of the regression analysis.

Figures and tables

Table 1: “Drinking” is not an appropriate expression to describe alcohol use. I recommend changing it to “History alcohol overconsumption” or similar – depending on what the authors used as criteria for alcohol intake (see previous comment under methods).

18% of the included patients had secondary osteoporosis. The validity of bone mineral density measurement in these patients is questionable, not necessarily reflecting fracture risk.

Author Response

Reviewer 2, Comment # 1:

Consider using the full name instead of the abbreviation for FRAX.

Response to Reviewer 2, Comment # 1:

We have added “Fracture Risk Assessment Tool” to the title.

Abstract:

Reviewer 2, Comment # 2:

The background and the objective of the study are not stated.

Response to Reviewer 2, Comment # 2:

We have added the following statements to the Abstract: “Trabecular bone score (TBS) was found to be significantly associated with moderate coronary artery calcification (CAC). The aim of this study was to further explore the association between TBS-adjusted Fracture Risk Assessment Tool (FRAX) and CAC score in women.” (line 16–18)

Reviewer 2, Comment # 3:

Page 1 line 29: “Increased risk of MOF and HF, determined by TBS-adjusted FRAX, is significantly and independently associated with CAC score in women.” – this is rather a repetition of the results that does not refer to the conclusion.

Response to Reviewer 2, Comment # 3:

We have revised the conclusion statement to “clinicians should be vigilant to the potential increased risk of coronary events among women with increased TBS-adjusted FRAX for MOF and HF.” following the suggestion by the reviewer. (line 29–31)

Keywords:

Reviewer 2, Comment # 4:

Page 1 line 31: “women” is a rather unspecific keyword, I recommend changing it.

Response to Reviewer 2, Comment # 4:

We have changed it to “adult women”. (line 32)

Introduction:

Reviewer 2, Comment # 5:

The description of the pathophysiological link between coronary artery calcification and bone mineral density could be more detailed. It is also somewhat unclear why TBS-adjusted FRAX would be a better predictor of CAC compared to TBS and FRAX alone, which was previously investigated by the authors.

Response to Reviewer 2, Comment # 5:

We have added description of the pathophysiological link between coronary artery calcification and bone mineral density in the Discussion section as follows:

“While the exact mechanism remains uncertain, several hypotheses have been proposed. Inhibitors of the Wnt signaling pathway, such as secreted frizzled Proteins 2 and 4 and Dickkopf-related protein-1, could play a role linking vascular calcification and bone loss [24]. Inflammatory cytokines and oxidized low density lipoproteins have also been suggested as determinants of vascular calcification and decrease in osteoblast activity. Moreover, the RANKL/RANK/OPG system and the cysteine protease cathepsin K were hypothesized to regulate vascular calcification and bone metabolism [25].” (line 219–226)

FRAX is widely used in clinical settings to assess the risk of fractures. The addition of TBS to FRAX might improve its accuracy to predict major osteoporotic fractures. In our previous study, we found that TBS was significantly associated with moderate but not high CAC in patients. However, the association between TBS-adjusted FRAX and CAC has not yet been explored. Therefore, the present study aimed to explore this association.

Materials and Methods:

Reviewer 2, Comment # 6:

The inclusion- and exclusion criteria are not described.

Response to Reviewer 2, Comment # 6:

We have revised the inclusion criteria in Section 2.1 as follows: women aged 20 to 80 years who had received both coronary computed tomography angiography and DXA scans at their general health examination. Exclusion criteria included the presence of metal implants in the measured body parts, or the possibility of pregnancy. (line 64–67)

Reviewer 2, Comment # 7:

The details of the coronary artery computed tomography (CT) image acquisition are not described (resolution, kV, etc.). It is not stated which images were used for Agatston-score analysis (e.g. original thin-slice images or multiparametric reconstruction images). It is also unclear which software was used for Agatston-scoring.

Response to Reviewer 2, Comment # 7:

We have revised the corresponding text according to the reviewer’s comment as follows:

“Unenhanced axial images (with tube voltage of 120 kVp, slice thickness at 2.5 mm, 512 matrix size, original thin-slice collimation, EKG-gating, breath holding) scanned before coronary computed angiography were used to obtain CAC scores. The scans were performed using a multidetector computed tomography system (LightSpeed VCT, GE Medical Systems, GE Healthcare, Chicago, IL, USA). The Agatston scoring method was used to quantify CAC [17]. Automated detection of calcific lesions was first processed on an Advantage Workstation, AW 4.3-09 (GE Healthcare, Chicago, IL, USA) and the exclusion of calcific lesions other than the coronary arteries (such as bones, lymph nodes, and lung lesions) was performed manually. A total calcium score was obtained by summing the individual scores of the left main, left anterior descending, circumflex, and right coronary arteries with their main branches.” (line 87–97)

Reviewer 2, Comment # 8:

The TBS and FRAX scores could be included in the descriptive analysis.

Response to Reviewer 2, Comment # 8:

We thank the reviewer for the suggestion. We have added summary statistics for TBS and FRAX scores in Table 1.

Reviewer 2, Comment # 9:

The time interval between the coronary artery CT and the dual-energy X-ray absorptiometry (DXA) scan is not described, which is important since longer times between the two scans could bias the results.

Response to Reviewer 2, Comment # 9:

We have added the following sentence in the revised manuscript: “Blood sampling, coronary computed angiography, and DXA scans for every patient were performed within a single general health examination.”. (line 84–85)

Reviewer 2, Comment # 10:

Page 2 line 64: the authors might mean “anthropometric” instead of “anthropomorphic” characteristics.

Response to Reviewer 2, Comment # 10:

We apologize for the typographical error. We have changed the term to “anthropometric” in the revised manuscript. (line 73)

Reviewer 2, Comment # 11:

Page 2 line 65-69: It is unclear when the listed laboratory parameters were obtained in relation to the date of the DXA scan and the coronary CT scan.

Response to Reviewer 2, Comment # 11:

We have added the following sentence in the revised manuscript: “Blood sampling, coronary computed angiography, and DXA scans for every patient were performed within a single general health examination.”. (line 84–85)

Reviewer 2, Comment # 12:

Page 2 line 70-71: The authors do not describe the definition of smoking status and alcohol use.

Response to Reviewer 2, Comment # 12:

We defined smoking status and alcohol use according to the FRAX tool, which are as follows:

Smoking defined as yes if the patient currently smokes tobacco.

Alcohol use defined as yes if the patient takes 3 or more units of alcohol daily. A unit of alcohol is approximately equal to a standard glass of beer (285 mL) or a single measure of spirits (30 mL). (line 80–84)

Reviewer 2, Comment # 13:

Page 2 line 88-94: This description does not belong to the methods and would rather fit in the introduction and discussion.

Response to Reviewer 2, Comment # 13:

We have moved the description of the trabecular bone score to the introduction section. (line 49–56)

Results

Reviewer 2, Comment # 14:

The words “associated” and “correlated” are frequently used in the results section. It is advised that the authors to also use “predicted/predictors” in order to adequately reflect the meaning of the regression analysis.

Response to Reviewer 2, Comment # 14:

We appreciate the suggestion from the reviewer. As our study was based a cross-sectional design, which is observational in nature, we prefer not to over-interpret the observed correlation as causality. We hope the reviewer will agree with our view.

Figures and tables

Reviewer 2, Comment # 15:

Table 1: “Drinking” is not an appropriate expression to describe alcohol use. I recommend changing it to “History alcohol overconsumption” or similar – depending on what the authors used as criteria for alcohol intake (see previous comment under methods).

Response to Reviewer 2, Comment # 15:

We appreciate the suggestion from the reviewer. We have changed the term to “alcohol use” in the tables.

Reviewer 2, Comment # 16:

18% of the included patients had secondary osteoporosis. The validity of bone mineral density measurement in these patients is questionable, not necessarily reflecting fracture risk.

Response to Reviewer 2, Comment # 16:

We agree that bone mineral density measurement alone does not necessarily reflecting fracture risk. Therefore, we have used the FRAX tool and the TBS-adjusted FRAX for the prediction of fracture risk.

Round 2

Reviewer 2 Report

Thank you for the responses of the authors and for the amendments in the manuscript. No further questions/comments are made.